# Orthotopically Implanted Murine Lung Adenocarcinoma Cell Lines for Preclinical Investigations

**DOI:** 10.3390/cancers17152424

**Published:** 2025-07-22

**Authors:** Karshana J. Kalyanaraman, Zachary Corey, Andre Navarro, Lynn E. Heasley, Raphael A. Nemenoff

**Affiliations:** 1Departments of Medicine, University of Colorado Anschutz Medical Campus, Aurora, CO 80045, USA; karshanakalyan@gmail.com (K.J.K.); andre.navarro@cuanschutz.edu (A.N.); 2Departments of Pathology, University of Colorado Anschutz Medical Campus, Aurora, CO 80045, USA; zachary.corey@cuanschutz.edu; 3Departments of Craniofacial Biology, University of Colorado Anschutz Medical Campus, Aurora, CO 80045, USA; 4Eastern Colorado VA Healthcare System, Rocky Mountain Regional VA Medical Center, Aurora, CO 80045, USA

**Keywords:** murine models, lung adenocarcinoma, orthotopic, ALK, EGFR, KRAS, RET

## Abstract

While new therapeutic approaches have improved the outcomes for patients with lung adenocarcinoma, there is great heterogeneity of response. Mouse models are critical in order to obtain a mechanistic understanding of the factors that drive this response and identify new therapeutic targets. This article reviews multiple models that have been used to study lung adenocarcinoma. The article focuses on the use of implantable orthotopic immunocompetent models using panels of murine lung cancer cell lines encompassing the drivers of human lung cancer. These models will be useful in developing and testing new therapeutic combinations to improve outcomes for lung cancer patients.

## 1. Introduction

Despite improved therapies, lung cancer remains the leading cause of cancer death in both men and women [1,2]. While new therapeutic approaches have been brought to the clinic, the majority of patients still eventually progress, underscoring the need for new therapeutic strategies. As a histologically defined subset of lung cancers, lung adenocarcinoma (LUAD^1^) is the most common primary lung cancer in the United States, representing ~40% of lung cancers. Importantly, LUAD frequently presents with druggable driver oncogenes, setting the stage for precision oncology with agents that specifically target distinct mutated receptor tyrosine kinases (RTKs) as well as mutated KRAS [3,4,5,6]. Moreover, immunotherapy using antibodies targeting the PD-1/PD-L1 immune checkpoint axis are approved first-line therapies in tumors that lack mutations in EGFR and ALK [7,8,9,10]. While unleashing host immunity is the target for anti-PD-1/PD-L1 inhibitors [11,12], the literature establishes that the tumor microenvironment (TME) and adaptive immunity are also critical contributors to the therapeutic efficacy of oncogene-targeted tyrosine kinase inhibitors (TKIs) and KRAS^G12C^ inhibitors [13,14,15].

Despite the significant advances provided by the application of precision oncology and checkpoint inhibitors in LUAD, progress towards long-term control or cures may have plateaued. We suggest that successfully addressing several outstanding questions may help overcome these therapeutic hurdles. The response to either targeted therapy or immunotherapy is heterogeneous, with oncogene-defined subsets of patients exhibiting wide variations in the depth and duration of response. The range in tumor shrinkage and time to progression are often depicted in “waterfall” and “swimmers” plots, respectively, where rare patients show a complete shrinkage, but the majority exhibit a partial response [3,16,17,18]. The mechanisms accounting for the variation in the depth and durability of therapeutic responses within an oncogene-defined subset of lung adenocarcinoma patients represents an important unanswered question in the field. Furthermore, the vast majority of LUAD patients treated with oncogene-targeted drugs develop resistance to therapy and undergo disease progression. For patients whose tumors present with on-target RTK mutations, third- and fourth-generation inhibitors have been developed [19], but bypass signaling still remains a major resistance mechanism [20,21,22,23]. Can mechanisms of acquired resistance to TKIs that lead to treatment failure be accurately anticipated such that next-generation combination therapies can be precisely deployed? As previously noted, the literature supports both positive and negative impacts of host immunity on the therapeutic response to oncogene-targeted drugs. What are the mechanisms involved and can they be intentionally manipulated for therapeutic gain? Finally, what are the molecular attributes of lung cancers that respond to immunotherapy versus those that are unresponsive? Why are LUADs driven by EGFR and ALK unresponsive to PD-1/PD-L1 inhibitors, yet apparently reliant on host adaptive immunity for a full clinical response?

How might these important and unanswered questions be best addressed? Certainly, state-of-the-art analyses of primary human tumor specimens for interactions between stromal cells within the TME, host immunity, and tumor cells, with a linkage to the degree of therapeutic responses, can provide insightful correlations. However, the mechanistic testing of specific signal pathways and immune cell subsets will require immunocompetent murine LUAD models [24]. While humanized mice have been developed that would allow the growth of human LUAD cells in vivo, these models are complex and have not been widely used [25], In vitro approaches using organoids have also been used to examine responses to therapy [26,27]. While these systems are relatively straightforward to develop, they do not incorporate all of the cell types of in vivo tumors, and the organization of cell populations is different. The utility of murine LUAD cell lines and the orthotopic implantation model for exploring the mechanisms by which communication between LUAD cancer cells and diverse cell types within the TME participate in therapeutic responsiveness is the focus of this article. We will briefly summarize the preclinical LUAD models that are currently employed. We will also review the murine LUAD cell lines that have been developed to date. Importantly, we will discuss the potential utility of immunocompetent implantable LUAD tumors, using panels of cell lines with distinct oncogenic drivers as a strategy to model the heterogeneity of response and identify critical interactions between cancer cells and the TME that may regulate variations in response and drug resistance. Since these models recapitulate critical features of human LUAD, they can identify new therapeutic targets and allow the testing of novel combination therapies that can be taken into clinical trials to improve the outcome for LUAD patients. For this article, we used the search terms lung adenocarcinoma, mouse models, therapeutic, and tumor microenvironment and focused on the last 10 years of publications. While this article considers mouse models, strategies using organoids or three-dimensional cultures provide alternatives to using mice. However, developing organoids encompassing all of the stromal cell types in the appropriate locations as models for in vivo tumors remains a significant obstacle to using these systems [27,28].

## 2. Chemical Carcinogen-Induced and Genetic Mouse Models

Approaches involving treating mice with chemical carcinogens such as urethane (vinyl carbamate) or NNK [29,30,31,32,33] to induce lung tumors have been employed for many decades. Mice usually develop multiple tumors in the lung, the majority of which are benign adenomas that eventually progress to form adenocarcinomas. In fact, the ability to induce lung tumors that progress from early to late stages of disease represents an advantage to this approach. The number of tumors is highly dependent on the strain of mice, with strains such as A/J developing a large number of tumors and C57BL/6 developing very few, even in the setting of multiple carcinogen administrations [32]. This has been attributed to inherent strain-related differences in immune cell populations [34]. In general, these carcinogens induce mutations in KRAS [35,36] and recapitulate the early stages of tumor development. Importantly, the spectrum of oncogenic drivers is limited to KRAS, and the tumors rarely undergo metastasis. Additional studies using genetically engineered mouse models (GEMMs) in which specific genes are deleted or overexpressed have been used to identify potential regulators of tumors’ initiation and progression [37,38,39]. For example, the targeted overexpression of prostacyclin synthase, the rate limiting enzyme in prostacyclin production in Type II pneumocytes, was shown to inhibit the initiation of carcinogen-induced tumors [40,41] and supported a chemoprevention trial using iloprost, a prostacyclin analog [42]. In contrast, the global deletion of a prostaglandin E2 receptor (EP2) protected mice against the formation of tumors [43]. However, it is difficult to define specific mechanisms involving communication between distinct cell types using carcinogen-induced systems, and this approach is not useful for examining the role of oncogenic drivers other than KRAS.

To better address the diversity of genetic drivers presented by LUAD, the past 20 years has seen the development of GEMMs in which specific oncogenic drivers identified in human LUAD are expressed in the lung. Originally, these models deployed specific oncogenic drivers that were doxycline-inducible or “knocked in” the appropriate oncogene using Cre-Lox technology. Initial studies expressing KRAS mutants demonstrated specific formation of lung tumors in response to the intratracheal administration of Adeno-Cre [44,45,46]. Studies by Tyler Jacks and others have used Cre/Lox-mediated recombination to generate the stable knock-in of mutant KRAS and simultaneously delete specific tumor suppressor genes such as p53 [44]. Genetic mice have also been developed for other oncogenic drivers, including EGFR [47,48], as well as ALK translocations [49]. These mice develop a more aggressive LUAD that can metastasize to distant organs and have been widely adopted for testing therapeutics. Moreover, the permanent expression of KRAS in tumors using the Cre-Lox system has led to the isolation of a large number of murine KRAS-mutant lung cancer cell lines (see below).

The strength of carcinogen-induced LUAD and GEMMs is that they recapitulate the development of tumors from preneoplastic lesions to full-fledged adenocarcinomas. However, there are several disadvantages that limit the potential of this approach in defining the critical determinants of therapeutic response. From a practical point of view, the emergence of lung cancer with carcinogens and in GEMMs can require long periods of time, requiring a significant effort and cost to administer therapeutic agents. Perhaps more importantly, there is a heterogeneity of response both between individual tumors within the same mouse and variations from mouse to mouse. The treatment of tumor-bearing mice, with either targeted therapies such as osimertinib for EGFR-mutant tumors or immunotherapies such as anti-PD-1 for KRAS-driven tumors, results in variable degrees of tumor shrinkage, with some tumors showing a strong response and other tumors showing a minimal response. While this heterogeneity of response recapitulates what is observed in patients, this model is not amenable to defining the specific mechanisms that dictate the depth and duration of response. We assume that these differences are due to inherent differences in the cancer cells within each tumor or alternatively, differences in interactions with the surrounding TME. However, the responses in each mouse will be different in the number of tumors and in the depth of response, making this model difficult to use for mechanistic studies. Therefore, to identify potential pathways that regulate the specific nature of response to therapeutic agents, other experimental approaches are required.

## 3. Implantable Models

Implantable models involve the inoculation of cancer cells directly into a mouse (see Figure 1). The injected cells form a localized primary tumor and in many cases metastasize to distant organs. A major advantage of this approach is that tumor growth occurs over a relatively short period of time (weeks compared to months for GEMM models) and is generally highly reproducible. Furthermore, these models have the added advantage that cancer cells can be engineered in vitro to overexpress or delete specific genes to test the effects of specific pathways in cancer cells on tumor growth and therapeutic response. This will be discussed in greater detail below. The limitation of this approach is that tumor growth is initiated by a “full-fledged” cancer cell, and therefore it does not reproduce the early stages of tumor development. Work originally initiated by Minna and Gazdar [50,51,52], and extended by many other investigators, used samples of human lung tumors and resulted in a large panel of human lung cancer cell lines [53]. These have been extremely useful in understanding cancer cell intrinsic pathways in vitro [54,55,56,57]. Studying tumor growth in vivo requires xenograft models where human cells are implanted into immunodeficient mice. These studies have been performed in a variety of immunodeficient strains, including nu/nu mice and *Rag1*^−/−^ mice [58], and lack some or all adaptive immune cell populations. Improved models include *Rag2*^−/−^; *Il2rg*^−/−^ mice that lack T cells, B cells, and NK cells [59]. In the era of immunotherapy but also oncogene-targeted agents, this is a major limitation since the tumors develop and are treated in the absence of an adaptive immune system. Therefore, interactions between cancer cells and the surrounding TME will not reproduce what occurs in human tumors, and the response to agents that target anti-tumor immunity, such as immune checkpoint inhibitors, cannot be studied. A long-term goal would be to develop humanized mice for these experiments, although there are still major technical limitations with pursuing this approach [60,61]. For the appropriate crosstalk with human cancer cells, all of the immune cell populations should be human, with limited contributions from mouse immune cells. While better models are being developed, these are not readily available [25]. An alternative approach is to develop cell murine lung cancer cells derived from purebred strains of mice and to reimplant these into syngeneic mice (Table 1).

### 3.1. Subcutaneous Implantation

The establishment of murine tumors subcutaneously in the murine flank remains a standard approach to assessing the in vivo activity of therapeutic agents. The main advantage of this method is the ease of injecting tumor cells and the ability to non-invasively monitor changes in tumor volume during the course of treatment or progression using digital calipers, allowing the accurate determinations of tumor growth over time. A significant disadvantage relevant to lung cancer is the many differences that likely exist in the TME within the subcutaneous space relative to the lung environment. For example, alveolar macrophages in the lung have been shown to contribute to tumors’ progression [73,74], but this population is not present in subcutaneous tumors. While additional studies are required to delineate specific differences in populations of innate immune and stromal cells, these limitations should be considered in assessing the relevance of these models to human lung cancer in light of the increasingly important role of the TME.

### 3.2. Orthotopic Implantation

To more accurately reproduce the environment in which lung tumors develop, a limited number of studies have employed orthotopic implantation methods. This can be achieved by the direct injection of cells into the lung or by injection via the tail vein where cancer cells initially seed in the lung. Our group has adopted previously reported methods [75,76] whereby murine lung cancer cell lines are directly inoculated into the left lung of mice. The technique yields single tumors that can be non-invasively monitored throughout the course of treatment by microcomputed tomography (μCT). Moreover, metastasis to the brain and liver has been observed with KRAS-driven cell lines [77]. An additional strength of this model is the establishment of the tumor within a TME relevant to lung cancer (see Figure 2). In support, Li et al. previously reported a significant enrichment in regulatory T cells (Tregs) in tumors propagated in the flank versus the orthotopic site [78]. Moreover, tumors derived from the injection of CMT167 cells into the left lung exhibited a high responsiveness to anti-PD-1 inhibitors, while subcutaneous flank tumors from the same cell line were much less sensitive.

### 3.3. Pulmonary Seeding Through Intravenous Injection

The dissemination of murine lung cancer cells via intravenous injection yielding multifocal lung tumors is a well-established method for assessing therapeutic responses in the pulmonary microenvironment [59,69,79]. Injected cells will largely form lung tumors but can also migrate through the circulation to other organs. Differences in immune regulation between pulmonary and subcutaneous tumors have been reported and related to differences in T cell priming that occur in the draining lymph nodes [80]. A technical issue with intravenous tumors relates to the complexity of quantification, since this method often yields multiple lung tumors. Due to the large number of cancers that form, quantification of the tumor burden using imaging techniques such as μCT or bioluminescence can prove technically difficult and subject to inconsistent measuring techniques.

## 4. Rationale for Murine LUAD Cell Lines

A major hurdle in adopting immunocompetent implantable models has been the limited availability of murine lung cancer cell lines that accurately represent the variation observed within oncogene-defined subsets of human LUAD. Several cell lines, including Lewis Lung Carcinoma (LLC) cells and CMT64/167, were derived from spontaneous lung tumors that developed in C57BL/6 mice [62,64]. Both of these cell lines were determined to have KRAS mutations, with LLC harboring a G12C mutation and CMT167 a G12V mutation. To expand the number of cell lines, a number of groups have used GEMM models and isolated cell lines from mouse lung tumors as outlined in Figure 1 [66,81]. Most of these have oncogenic KRAS mutations and often have a loss of specific tumor suppressors, including p53 or LKB1 [59,68,82]. Importantly, these cells can be implanted into either wild-type or genetically altered C57BL/6 mice, allowing for a syngeneic tumor microenvironment.

While these cell lines have been useful representatives of the KRAS mutant subset of LUAD, a major advance in the treatment of LUAD has been the advent of personalized medicine focusing on specific oncogenic RTK drivers. At present, LUAD patients are subdivided according to positivity for specific mutations in KRAS and EGFR, as well as gene rearrangements yielding ALK, RET, NTRK, and ROS1 fusions. These tumors are treated with specific small-molecule agents that directly target the oncogenic RTK [5,16,83,84,85,86]. Importantly, the role of the immune system is dependent on the oncogenic driver. For example, while tumors expressing oncogenic KRAS show a strong response to immune checkpoint inhibitors, either as a monotherapy or in combination with chemotherapy, tumors driven by mutations or fusions in tyrosine kinase receptors show a poor response to this therapy [87]. Therefore, a mechanistic understanding of the interactions of cancer cells with the immune system using implantation models will require a panel of murine cells that reflects the major oncogenic drivers of human LUAD.

Using this approach, a panel of murine lung cancer cell lines has been developed that expresses the major drivers of human LUAD: KRAS (both G12D and G12C mutants), EGFR mutations, EML4-ALK fusions, and RET fusions (see Table 1). An important consideration in these cell lines is co-mutations, particularly in tumor suppressor genes such as p53. Another critical issue in isolating these cell lines is the strains of mice in which the tumors developed. Many GEMMs have been developed in mixed strains of mice, such as C57BL/6:129 hybrids. While cell lines derived from these mice are valuable to study cell-autonomous effects, they are not useful for examining the role of the adaptive immune system in pure mouse strains. The implantation of these cells into C57BL/6 mice initially results in tumor growth followed by, in many cases, immune elimination after 7–10 days. A subset of these lines can result in longer-term tumor growth, although there are significant challenges in studying the immune microenvironment in mixed mouse strains. Therefore, as a general strategy, deriving cell lines from tumors grown in pure strains of mice and using these cells to establish orthotopic tumors in the syngeneic host is the preferred approach.

### 4.1. Models for KRAS-Driven LUAD

Panels of murine LUAD cell lines expressing oncogenic KRAS have been developed by multiple groups of investigators [63,66,88]. Tumors established with these cell lines have been examined for their responsiveness to immune checkpoint inhibitors such as anti-PD-1, and more recently to specific KRAS inhibitors [59,63,65,85]. Most of these tumors are resistant to anti-PD-1 therapy, and studies have focused on both cancer cell intrinsic and extrinsic mechanisms that can modify the response to immunotherapy. These models have also proven useful in defining mechanisms of resistance to KRAS inhibitors.

### 4.2. Murine Models and Cell Lines for Oncogenic RTK-Driven LUAD

#### 4.2.1. Mutant EGFR

Multiple groups have developed GEMMs in which human EGFR cDNAs bearing the common oncogenic mutations (L858R or exon 19 deletions) are expressed in relevant lung cell types via tetracycline-inducible promoters to yield multifocal lung tumors [48,58,89,90,91]. While these models provided evidence for the therapeutic activity of TKIs and antibody-based agents, cultured cell lines have not been generated. Our groups [72] developed a GEMM where the murine EGFR cDNAs encoding the L860R (equivalent to human L858R) and exon 19 deletion mutations are controlled with a Lox-stop-Lox sequence such that the instillation of adenoviruses encoding Cre recombinase yields lung tumors. Multiple cell lines were derived that can be readily transplanted into the lungs of syngeneic mice to yield tumors that are highly responsive to osimertinib.

#### 4.2.2. ALK-Driven Tumors

Maddalo et al. [49] developed a CRISPR/Cas9 method to initiate EML4 and ALK gene rearrangements resulting in oncogenic EML4-ALK fusions in murine lungs following the instillation of engineered adenoviruses. The resulting tumors were sensitive to ALK inhibitors, and these investigators derived a cell line from these mice. More recently, we have expanded this approach to establish additional ALK+ cell lines that can be orthotopically transplanted into syngeneic C57BL/6 hosts [71]. Validation of the expression of the fusion protein and the sensitivity of these cells to ALK inhibitors, such as alectinib and lorlatinib, in vitro confirm that these cells are dependent on ALK signaling. All of the cell lines showed a similar growth inhibition in vitro. However, when these cells were implanted as orthotopic lung tumors into syngeneic mice, differences in the depth of response to alectinib were observed; one cell line showed a complete response with no detectable residual tumor cells, and the cessation of therapy did not lead to the regrowth of the tumor. Two other cell lines showed a partial response, with the presence of a stable residual disease, and tumors regrew rapidly upon stopping alectinib treatment. Thus, this panel of cell lines demonstrates the heterogeneity of response seen in cohorts of human ALK+ patients [92]. Interestingly, the implantation of identical cells into immunodeficient mice resulted in only a transient growth inhibition, with all cell lines progressing even in the setting of continued alectinib treatment, indicating that adaptive immunity contributes to the durability of TKI responses.

#### 4.2.3. RET-Driven Lung Tumors

Beyond EGFR and ALK, chromosomal rearrangements yielding oncogenic forms of ROS1, RET, and NTRK family members now represent subsets of LUAD associated with FDA-approved TKIs [5,6]. The availability of murine cell lines that reflect these subsets would permit an in vivo evaluation of experimental therapeutics and mechanisms of acquired resistance. Resistance can be mediated by additional somatic mutations within the RTK coding sequences but also often involves the induction of bypass signaling pathways [6,19]. To this end, Schubert et al. developed a novel in vitro screening strategy using Ba/F3 cells to identify pairs of single guide RNAs (sgRNAs) that successfully generate chromosomal rearrangements in murine RET (*Kif5b-Ret*, *Trim24-Ret*) and NTRK1 (*Tpm3-Ntrk1*) genes [93]. These studies set the stage for the development of adenovirus constructs containing the validated sgRNAs in immune-competent mice for the purpose of establishing murine equivalents for these oncogene-defined LUAD subsets. Two cell lines have been derived from CRISPR-Cas9-induced *Trim24-Ret* lung tumors [93,94] and are distinct from EGFR and ALK+ cell lines, yielding a stable residual disease upon TKI treatment [71]; the *Trim24-Ret* orthotopic tumors initially respond to the TKI, selpercatinib, and then promptly progress on continued treatment. These murine RET+ cell lines thus open avenues to investigate in vivo mechanisms of acquired resistance and optimized combination therapies.

## 5. Critical Issues in LUAD Therapeutics That Can Be Addressed with Orthotopic Implantable Murine Models

In this section we will address the aforementioned critical issues, where the deployment of panels of murine LUAD cell lines and orthotopic immunocompetent models may provide novel insights.

### 5.1. What Are the Molecular Attributes of Tumors That Respond to Immunotherapy Versus Those That Are Unresponsive?

These issues can be defined using panels of murine cancer cell lines that show reproducible differences in the response to immunotherapy agents such as anti-PD-1 antibodies. The panel of KRAS-driven cell lines replicates these responses, with orthotopic CMT167 tumors showing a complete response, LLC tumors being resistant, and mKRC.2 showing a partial response [63]. Our previous studies have defined the features of these cancer cells that contribute to the response. The expression of MHC class II on cancer cells is associated with an improved response to immune checkpoint inhibitors in melanoma [95]. CMT167 tumors also manifest an expression of MHC class II on cancer cells in vivo [96]. To test the role of this pathway, CMT167 cells lacking MHC class II were generated using shRNA against CIITA, the master regulator of MHC II. Implanting these cells in the orthotopic model revealed that the loss of MHCII resulted in an impaired response to anti-PD-1 therapy, determining that MHCII, specifically, had an instrumental role in the cancer cells’ response to immunotherapy. On the other hand, the induction of interferon signaling has been associated with an improved response to anti-PD-1 therapy [97]. An analysis of our murine lung cancer cell lines demonstrated that IFN signaling was less robust in the resistant LLC cells, compared to the sensitive CMT167 cells [98]. An examination of RNA-seq data indicated that LLC had higher levels of SOCS1 expression, an endogenous inhibitor of JAK/STAT signaling. Silencing SOCS1 increased the sensitivity of cancer cells to IFNγ stimulation and improved the response to anti-PD-1 antibodies. These models can also be used to define differences in response to KRAS^G12C^ inhibitors [63,65,79]. The characterization of populations of immune and inflammatory cells using these models has identified potential targets in the TME to improve therapeutic responses [65,99,100].

### 5.2. Why Is LUAD Driven by EGFR and ALK Unresponsive to PD-1/PD-L1 Inhibitors?

While the development of immune checkpoint inhibitors has been a major advance in the treatment of LUAD, patients with mutations in EGFR or ALK fusions show a very low response rate to these therapies, and there are significant toxicities [87]. While some investigators have concluded that this is due to the lack of an inflamed microenvironment or a low mutation burden in these subsets of LUAD [101], multiple studies have demonstrated an important role for adaptive immunity in mediating the response to TKIs [20]. Thus, defining how adaptive immunity productively contributes to TKI responsiveness, but not immune checkpoint inhibitor sensitivity, in these LUAD subsets and identifying novel targets to engage the immune system is a critical unmet need for these patients. Relevant murine models will be critical to identifying these targets. Consistent with what is observed in patients, the implantable models of either EGFR- or ALK-driven lung cancers show no response to anti-PD-1 therapy but show a clear role for adaptive immunity in the response to TKIs [71]. Therefore, these cell lines represent an excellent system for defining the factors that modulate anti-tumor immunity.

Why do oncogene-defined subsets of LUAD exhibit a wide range of therapeutic responses to TKIs defined by the degree of initial tumor shrinkage and duration of the response? As discussed above, panels of oncogene-defined LUAD cell lines that show similar responses to TKI in vitro show varying responses in orthotopic implantation that resemble the waterfall plot observed in clinical trials with TKIs. Using these models, one can begin to define critical features on the cancer cells that may mediate this response. Our group has performed an RNA-seq analysis of cancer cells recovered from tumors to begin to define these critical differences. For example, EA2, an ALK-driven cell line that shows a complete response to TKI treatment, produces high levels of CXCL9 and CXCL10 in vivo [71]. This is associated with an increased CD8 T cell infiltration, consistent with an anti-tumoral role for this immune population. Conversely, EA1 tumors that show a partial TKI response, with significant residual disease, produce low levels of these chemokines and consequently have lower numbers of infiltrating T cells. In addition, EA1 tumors produce higher levels of CXCL1/2, chemokines associated with the recruitment of immunosuppressive neutrophils. Importantly, the orthotopic model allows a mechanistic assessment of the importance of these factors. For example, cells can be engineered using either shRNA or CRISPR approaches to delete specific chemokine genes. The knockout cells can be implanted into mice and effects on tumor growth and response to TKI therapy examined. These studies can be further extended to test pharmacologic agents such as inhibitors of the CXCR2 or CXCR3 receptors. The power of the implantable models resides in the ability to rapidly engineer cancer cells and use the methodologies developed to define how the cell-specific deletion affects tumor growth.

Another case where these models have the power to provide mechanistic insights is in examining the role of lipid mediators that lead to immunosuppression. Recent studies have shown that prostaglandin E2 (PGE_2_) mediates immunosuppression by inhibiting T cells, NK cells, and dendritic cells [59]. However, PGE_2_ can be produced by numerous cells within the TME, including macrophages and fibroblasts, as well as cancer cells. Defining the relative importance of PGE_2_ production by each cell type, as well as examining other potential lipid mediators, remains poorly defined. With implantable LUAD cell lines and the orthotopic model, enzymes mediating PGE_2_ production, such as cyclooxygenase 1 and 2 and prostaglandin E synthase, can be deleted in cells that express them or alternatively be overexpressed in cells that lack expression. These engineered cells can then be used to selectively test the role of cancer cell-derived PGE_2_ on tumor growth. Conversely, implanting parental cancer cells into mice that have been engineered to knock out these enzymes in specific populations such as macrophages will allow an assessment of the relative contribution of these cells to the overall response.

### 5.3. Mechanisms of Acquired Resistance to Oncogene-Targeted Agents in an Immune-Competent Microenvironment

Implantable immunocompetent models can be used to define and test specific pathways that mediate acquired resistance. While GEMMs have been used to examine resistance pathways, the reproducibility of the response of implanted tumors to either targeted therapies or immunotherapies allows mechanistic insights to be tested. There are several strategies that can be employed to generate cancer cells that are resistant to therapy. Murine cancer cells grown in vitro for longer periods in low concentrations of specific inhibitors result in the outgrowth of cells that proliferate even in the presence of the specific agent [102]. These cells can be characterized for pathways of resistance and implanted into mice treated with the specific agent. Alternatively, tumors that progress in the setting of treatment with either targeted agents or immunotherapy can be isolated, and additional cell lines derived. Once potential pathways of resistance have been defined, murine cancer cells can be engineered to target these pathways and orthotopic tumors tested for increased sensitivity to therapeutics. As a proof of concept, Molina-Arcas et al. previously reported that LLC cells contained an NRAS-Q61H gain-of function mutation in addition to a KRAS^G12C^ mutation [65]. Engineering LLC cells to delete NRAS increased the response to KRAS-G12C inhibitors in orthotopic mouse tumors [63,65]. Similarly, our group has shown that the cancer cell expression of MHC class II is associated with a strong response to anti-PD-1. Silencing the expression of CIITA, the master regulator of MHC Class II, in cancer cells that express MHC II in vivo blunted the response to anti-PD-1, whereas expressing CIITA in resistant cells sensitized tumors to anti-PD-1 therapy [96]. Finally, our recent studies [94] with murine Trim24-Ret cell lines provide an example of acquired resistance to RET-targeting TKIs and an assessment of the efficacy of combining MET-specific TKIs.

## 6. Challenges and Limitations

The development of a panel of murine LUAD cell lines encompassing the major oncogenic drivers of human LUAD will provide an important resource for addressing the various hurdles that have emerged in the application of precision oncology. While the panels of cells developed are useful in developing mechanistic insights, there are a number of technical and scientific limitations that need to be addressed. While a limited number of cell lines have been developed, they clearly do not nearly represent the numbers of human cell lines developed over many years. Even within specific oncogenic driver-defined subsets, there are insufficient cell lines to assess the role of specific mutations or the co-expression of other classes of genes such as tumor suppressor genes. For example, the EML4-ALK cell lines developed by our lab are all derived from the V1 variants, and there are no cell lines expressing the V3 variant. In addition, there are not adequate cell lines to assess the role of tumor suppressors such as p53 on response to therapy. Of the three ALK cell lines, two are p53-null and one is p53 WT. Similar concerns exist regarding the EGFR-dependent cell lines that have been developed. To develop a more comprehensive panel will require extensive work generating novel GEMMs that can then be used to develop cell lines.

In addition, it is not clear how many implantable cell lines will be required to accurately reflect the range in responses to therapy observed in oncogene-defined subsets of LUAD. For example, as discussed above, with just three ALK+ cell lines we have observed a range in the depth of response to alectinib. However, all of the mutant EGFR cell lines developed show a similar complete response to osimertinib, although additional lines are being analyzed. Thus, to reproduce the “waterfall plot” seen in patients will require significantly more cell lines. Again, the role of specific co-mutations or mutations in tumor suppressor genes also needs to be addressed.

Finally, it is likely that implantable models lack the intratumoral heterogeneity observed in advanced lung cancers diagnosed in patients [103,104]. While some of the murine cancer cells we have derived from GEMM tumors are not clonal, they certainly do not have the complexities of human tumors. There are potential strategies using implantable orthotopic tumors that can be used to address the role of intratumoral heterogeneity in mediating the response to therapy. One approach is to inject mixtures of different cell lines bearing the same oncogene but exhibiting distinct TKI responses. This has been used in subcutaneous injections of mixtures of carcinogen-derived squamous skin carcinoma cells [105]. In this study, cells that formed T cell-rich (hot) tumors or T cell-poor (cold) tumors were mixed together to generate more complex tumors. Each cell population was labeled with specific fluorophores to allow identification by fluorescence. This approach can readily be applied to orthotopic lung implantation models, using cells with the same oncogenic drivers but differing in co-mutations or showing differences in response.

## 7. Conclusions

Research over the past 50 years has generated a large number of mouse models to study lung cancer [28,58,60,61,106] and all have strengths and limitations. Genetic mouse models and chemical carcinogenic models are critical in developing an understanding of tumors’ evolution and the transition from premalignancy to full-fledged cancer. This review has focused on implantable immunocompetent models, but several other reviews have examined GEMM models and organoid approaches [24,28,60,107]. While these models show heterogeneous responses to therapy, these models are difficult to use to identify mechanistic targets. As a complementary approach, the recent development of orthotopic immunocompetent implantable models is a powerful tool to model the variation in response to approved targeted therapies observed in patients expressing the same oncogenic driver. The reproducibility of response seen with the same cell line implanted into multiple mice enables mechanistic studies to define pathways both in the cancer cell and in the TME that mediate the depth of response, time to progression, and acquired resistance. Critical to the long-term goal to identify novel therapeutic targets is the development of a panel of cell lines that accurately reflect the differences in response. The role of co-mutations and loss of specific tumor suppressors also needs to be considered. This may require generating a large number of cell lines. However, the power of this model to mimic human lung cancer and to define mechanisms of response makes this an invaluable asset in improving the outcome for lung cancer patients.

## Figures and Tables

**Figure 1 cancers-17-02424-f001:**
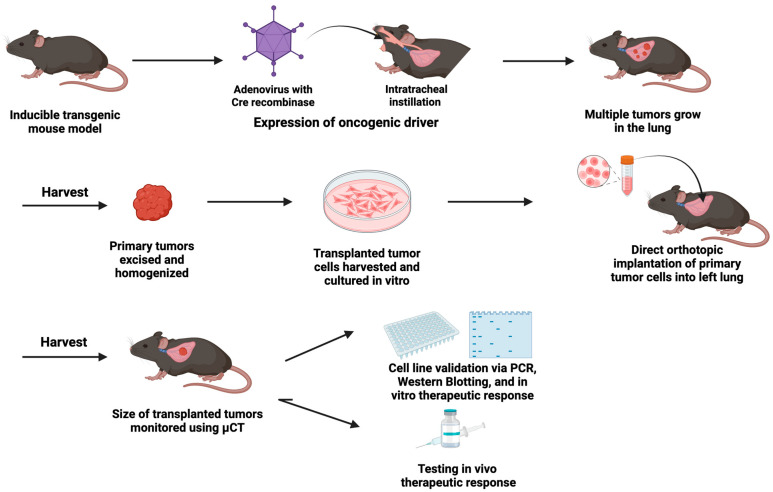
Derivation of Murine Lung Cancer cell lines from GEMMs. **A**. The administration of adenovirus encoding Cre recombinase via intratracheal administration results in the “knock in” of oncogenic drivers in inducible transgenic mouse models. This results in oncogene expression and the development of multiple lung tumors. **B**. These tumors can be excised and placed into a culture to derive stable murine lung cancer cell lines. Cancer cells are validated for the expression of the oncogenic driver and in vitro growth inhibition by targeted inhibitors such as TKIs. **C**. Cancer cells can be reimplanted into syngeneic mice and the resulting tumors measured non-invasively using µCT. Mice can be treated with therapeutic agents, either targeted small molecule inhibitors or immunotherapy such as anti-PD-1, and changes in tumor size determined as a function of time using µCT.

**Figure 2 cancers-17-02424-f002:**
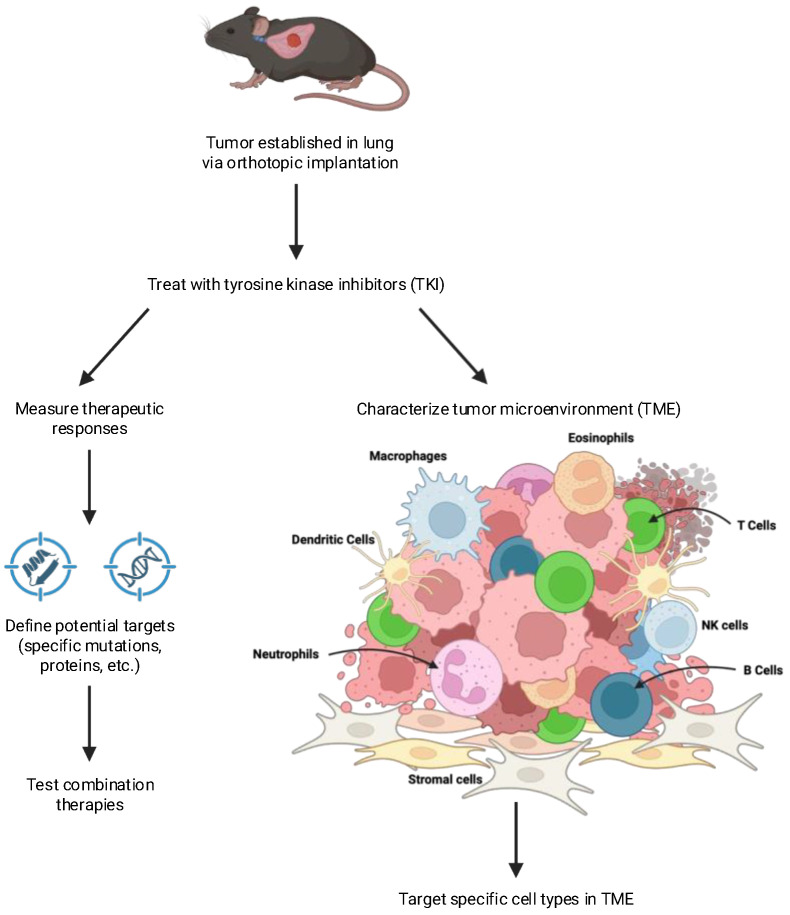
Defining changes in tumors in response to therapy. Orthotopic tumors implanted into syngeneic mice can be used to identify novel therapeutic targets. Cancer cells can be recovered from control or treated tumors and changes in gene expression determined. A comparison of complete responders vs. partial responders having residual disease can be used to identify potential targets that can be tested with combination therapies. Changes in the tumor microenvironment in response to therapy can be determined using methods such as flow cytometry, multispectral imaging, and spatial transcriptomics. Comparison of the TME from tumors showing different depths of response can be used to highlight specific agents that can target distinct cell types within the TME.

**Table 1 cancers-17-02424-t001:** Oncogene-defined murine lung cancer cell lines.

	Cell Line	Strain Derivation	Driver Oncogene	Co-Mutations	Therapies Tested	Reference
KRAS						
	CMT167	C57BL/6	KRAS-G12V		anti-PD1/PD-L1	[62]
	CMT-G12C	C57BL/6	KRAS-G12C			[63]
	LLC	C57BL/6	KRAS-G12C	NRAS-Q61H; TP53-R334P	anti-PD1/PD-L1	[64]
	LLC NRAS KO	C57BL/6	KRAS-G12C	TP53-R334P	MRTX-849 −/+ RMC-4550	[63]
	3LL NRAS KO	C57BL/6	KRAS-G12C	TP53-R334P		[65]
	mKRC.1	C57BL/6	KRAS-G12C	TP53^−/−^	MRTX-849 −/+ RMC-4550	[63]
	393P, 393LN, 344SQ	129Sv	KRAS	TP53^R172HDg/+^	Role of EMT	[66]
	KPAR1 and derivatives		KRAS-G12D		Anti-PD1/KRAS-G12C inhibitors	[59]
	CT-26 KRAS G12C		KRAS-G12C		KRAS G12C inhibitor	[67]
	KPAR G12C		KRAS-G12C		KRAS G12C inhibitor	[68]
	KP lines	C57BL/6-129sV	KRAS-G12D	TP53^−/−^		[69]
	mTC11	C57BL/6-129sV	KRAS-G12D		Anti-PD-1	[70]
	mTC14	C57BL/6-129sV	KRAS-G12D		Anti-PD-1	[70]
**ALK**						
	EA1	C57BL/6	Eml4-Alk		alectinib	[71]
	EA2	C57BL/6	Eml4-Alk	TP53^−/−^	alectinib	[49]
	EA3	C57BL/6	Eml4-Alk	TP53^−/−^	alectinib	[71]
**EGFR**						
	mEGFRdel19.1	C57BL/6	Egfr exon19 del	TP53^−/−^	osimertinib	[72]
	mEGFRdel19.2	C57BL/6	Egfr exon19 del	TP53^−/−^	osimertinib	[72]
	mEGFRL860R.1	C57BL/6	Egfr-L860R	TP53^−/−^	osimertinib	[72]

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
