# Peer review of "Orthotopically Implanted Murine Lung Adenocarcinoma Cell Lines for Preclinical Investigations"

_cancers, 2025, doi:10.3390/cancers17152424_

Round 1
Reviewer 1 Report
Comments and Suggestions for Authors
Dear Authors,
It is thought that the subject of the study is completely compatible with your journal. I believe that studies on this subject and similar one will contribute to the academy. I believe that article can be published after corrections. Suggested corrections;
Please start the abstract with a capital letter.
Explain the necessity of the study more clearly at the beginning of the introduction section, taking into account the increasing number of cancer cases and the resulting deaths.
Please add the abbreviations you use in the article in full text at their first occurrence.
At the end of the introduction, please explain the necessity and importance of your work in more detail.
At the end of the introduction, please also include a detailed description of the stages of your work and how many publications on different topics you have reviewed.
Compare the studies conducted in the literature and explain the shortcomings of these studies in more detail.
Please comment further on what can be done in future publications, especially considering the explanations of these studies in the conclusion section.
The contributions of this and similar studies should be included in the conclusion section.
Yours Sincerely
Author Response
Reviewer #1:
Dear Authors,
It is thought that the subject of the study is completely compatible with your journal. I believe that studies on this subject and similar one will contribute to the academy. I believe that article can be published after corrections. Suggested corrections;
Please start the abstract with a capital letter.
We have made that correction.
Explain the necessity of the study more clearly at the beginning of the introduction section, taking into account the increasing number of cancer cases and the resulting deaths.
We have added a sentence to emphasize the gaps in knowledge to be addressed by the article.
Please add the abbreviations you use in the article in full text at their first occurrence.
We have added a footnote containing all the abbreviations which is referenced at the first occurrence.
At the end of the introduction, please explain the necessity and importance of your work in more detail.
We have added text at the end of the introduction to clarify the importance of the studies.
At the end of the introduction, please also include a detailed description of the stages of your work and how many publications on different topics you have reviewed.
We have described the search terms used with either Pubmed or Google Scholar to survey the literature within the last 10 years. This has been added to the end of the introduction.
Compare the studies conducted in the literature and explain the shortcomings of these studies in more detail.
We have added a section in the Conclusions
Please comment further on what can be done in future publications, especially considering the explanations of these studies in the conclusion section.
We have discussed this in the section on challenges and limitations.
The contributions of this and similar studies should be included in the conclusion section.
We have added citations to other reviews in this field to the end of the conclusions.
Reviewer 2 Report
Comments and Suggestions for Authors
The proposed manuscript "Orthotopically Implanted Murine Lung Adenocarcinoma Cell Lines for Preclinical Investigations" appear as a synthetic, narrative review on a well-focused topic.
It could be of great interest for researchers aiming to choose the appropriate animal model to address issues related to better understanding the pathogenetic mechanisms of LUAD and to develop new treatment strategies and/or overcome drug resistance.
Overall, in my opinion, some themes have been addressed and could be promoted more efficiently by carefully connecting the logical concatenation of the reported scientific questions/answers, while other topics could be integrated to further improve the manuscript.
Below, I list the minor concerns and some aspects that I would suggest the authors would review:
1) Please check that after you define an abbreviation for the first time (e.g. page 1, line 34), you use the acronym regularly (e.g. page 2, lines 54-58; page 4, line 154, page 10, line 403: "GEMM models"- "GEMMS"?).
2) Please check through the manuscript for typos (e.g. page 2, line 86-88: "we will discuss the potential utility of immunocompetent - murine models of orthotopic LUAD?- implantable tumors using panels of cell lines with distinct oncogenic driver....; e.g. page 3, line 122: Genetic mice -models?- or use acronim GEMMs; e.g. page 8, line 326 "T"-he ?; page 11, line 473: "..with the same cell line implanted into multiple mice- into different mouse models?- enables...").
3) I would suggest improving their review by detailing the search strategy, time frame of reference, mesh terms, and search database used to balance information from both self-citations and other authors' articles.
4) Abstract: "In this review we focus on preclinical models that can be used to explore these interactions, identify new therapeutic targets, and test combination therapies. In particular, we will describe the use of implantable orthotopic immunocompetent models employing a panel of cell lines with oncogenic drivers common to human lung adenocarcinoma as a powerful system to develop new combination therapies". Here, you could outline your specific, extensive expertise to introduce the description of your studies, as well as refer to reading other bibliography (XX) to find details on models other than orthotopically implanted murine Lung Adenocarcinoma Cell Lines.
5) Page 2, line 75-79: I would suggest explaining in more detail the strengths and limitations of humanized mouse models and in vitro alternatives (e.g. organoids) before introducing the animal model you will primarily describe. Some hypotheses are mentioned in specific paragraphs (e.g. page 4, line 140-146), but I would suggest supporting them with relative references. In addition, you should mention somewhere in the manuscript why it is not currently possible to replace animal models of LUAD and the ethical concerns related to those currently available and indicated, including refinement of microsurgical methods to generate orthotopic LUAD mouse models (e.g. ultrasound-guided implantation) and the animal care required for the proper use of these models from both a humane and scientific standpoint.
6) In several parts of the manuscript, the role of the interaction between host immunity, TME and implanted tumor cells is mentioned, as well as the mouse strain used in the cited references (page 6, lines 248-258; Table 1). However, I would suggest addressing in more detail and clearly the importance of choosing the strain of mice to use for these purposes, also reporting your experience in this regard. I suggest to emphasize that the "model system" is constituted by the correct combination of "mouse model + cell line" (page 9, line 362-363), as well as to carefully describe the advantages and limitations of LUAD cell implantation (xenograft/orthotopic) in immunocompromised/syngeneic mouse models.
7) In several parts of the manuscript, the potential emergence of resistance to oncotherapeutics in relation to a specific genetic mutation is mentioned.
Could the authors address in more detail the specific mechanisms involved in the cases cited?
Author Response
Reviewer #2:
The proposed manuscript "Orthotopically Implanted Murine Lung Adenocarcinoma Cell Lines for Preclinical Investigations" appear as a synthetic, narrative review on a well-focused topic.
It could be of great interest for researchers aiming to choose the appropriate animal model to address issues related to better understanding the pathogenetic mechanisms of LUAD and to develop new treatment strategies and/or overcome drug resistance.
Overall, in my opinion, some themes have been addressed and could be promoted more efficiently by carefully connecting the logical concatenation of the reported scientific questions/answers, while other topics could be integrated to further improve the manuscript.
Below, I list the minor concerns and some aspects that I would suggest the authors would review:
1) Please check that after you define an abbreviation for the first time (e.g. page 1, line 34), you use the acronym regularly (e.g. page 2, lines 54-58; page 4, line 154, page 10, line 403: "GEMM models"- "GEMMS"?).
We have made that correction.
2) Please check through the manuscript for typos (e.g. page 2, line 86-88: "we will discuss the potential utility of immunocompetent - murine models of orthotopic LUAD?- implantable tumors using panels of cell lines with distinct oncogenic driver....; e.g. page 3, line 122: Genetic mice -models?- or use acronim GEMMs; e.g. page 8, line 326 "T"-he ?; page 11, line 473: "..with the same cell line implanted into multiple mice- into different mouse models?- enables...").
We have edited the manuscript for typos, and clarified sections.
3) I would suggest improving their review by detailing the search strategy, time frame of reference, mesh terms, and search database used to balance information from both self-citations and other authors' articles.
We have added information regarding our search strategy at the end of the introduction.
4) Abstract: "In this review we focus on preclinical models that can be used to explore these interactions, identify new therapeutic targets, and test combination therapies. In particular, we will describe the use of implantable orthotopic immunocompetent models employing a panel of cell lines with oncogenic drivers common to human lung adenocarcinoma as a powerful system to develop new combination therapies". Here, you could outline your specific, extensive expertise to introduce the description of your studies, as well as refer to reading other bibliography (XX) to find details on models other than orthotopically implanted murine Lung Adenocarcinoma Cell Lines.
We have incorporated information regarding our search strategy at the end of the introduction.
5) Page 2, line 75-79: I would suggest explaining in more detail the strengths and limitations of humanized mouse models and in vitro alternatives (e.g. organoids) before introducing the animal model you will primarily describe. Some hypotheses are mentioned in specific paragraphs (e.g. page 4, line 140-146), but I would suggest supporting them with relative references. In addition, you should mention somewhere in the manuscript why it is not currently possible to replace animal models of LUAD and the ethical concerns related to those currently available and indicated, including refinement of microsurgical methods to generate orthotopic LUAD mouse models (e.g. ultrasound-guided implantation) and the animal care required for the proper use of these models from both a humane and scientific standpoint.
We have added a discussion of organoid models in the introduction, and have expanded the discussion of humanized mice, as suggested. We have also added new references on these topics.
6) In several parts of the manuscript, the role of the interaction between host immunity, TME and implanted tumor cells is mentioned, as well as the mouse strain used in the cited references (page 6, lines 248-258; Table 1). However, I would suggest addressing in more detail and clearly the importance of choosing the strain of mice to use for these purposes, also reporting your experience in this regard. I suggest to emphasize that the "model system" is constituted by the correct combination of "mouse model + cell line" (page 9, line 362-363), as well as to carefully describe the advantages and limitations of LUAD cell implantation (xenograft/orthotopic) in immunocompromised/syngeneic mouse models.
We have expanded this discussion on page 11
7) In several parts of the manuscript, the potential emergence of resistance to oncotherapeutics in relation to a specific genetic mutation is mentioned. Could the authors address in more detail the specific mechanisms involved in the cases cited?
We have addressed this on page 13 and added additional references.
Reviewer 3 Report
Comments and Suggestions for Authors
The manuscript by Kalyanaraman et al. describes the use of implantable orthotopic immunocompetent models that utilize a panel of cell lines with oncogenic drivers common to human lung adenocarcinoma, serving as a robust system for the development of new combination therapies.
I have a few comments for polishing the manuscript.
- The authors are encouraged to include figures that illustrate the implantable model systems.
- The authors have examined chemical carcinogen-induced and genetic mouse models, as well as implantable models. It is essential to evaluate the advantages and disadvantages of one process in relation to another.
- There are some repetitions of text throughout the manuscript. The authors should avoid that.
Author Response
Reviewer #3:
The manuscript by Kalyanaraman et al. describes the use of implantable orthotopic immunocompetent models that utilize a panel of cell lines with oncogenic drivers common to human lung adenocarcinoma, serving as a robust system for the development of new combination therapies.
I have a few comments for polishing the manuscript.
The authors are encouraged to include figures that illustrate the implantable model systems.
Figure 1 in the review described the implantable model.
The authors have examined chemical carcinogen-induced and genetic mouse models, as well as implantable models. It is essential to evaluate the advantages and disadvantages of one process in relation to another.
We have presented advantages and disadvantages of chemical carcinogen-induced lung tumor models relative to implantation of murine lung cancer cell lines derived from GEMMs on page 6 of the manuscript.
There are some repetitions of text throughout the manuscript. The authors should avoid that.
We have tried to remove redundancies in the text.
Round 2
Reviewer 3 Report
Comments and Suggestions for Authors
The manuscript by Kalyanaraman et al. describes the use of implantable orthotopic immunocompetent models that utilize a panel of cell lines with oncogenic drivers common to human lung adenocarcinoma, serving as a robust system for the development of new combination therapies.
The authors have addressed all the previous comments. Thus, the manuscript can be accepted in its present form.